# Disruption of Mitophagy Flux through the PARL-PINK1 Pathway by CHCHD10 Mutations or CHCHD10 Depletion

**DOI:** 10.3390/cells12242781

**Published:** 2023-12-07

**Authors:** Tian Liu, Liam Wetzel, Zexi Zhu, Pavan Kumaraguru, Viraj Gorthi, Yan Yan, Mohammed Zaheen Bukhari, Aizara Ermekbaeva, Hanna Jeon, Teresa R. Kee, Jung-A Alexa Woo, David E. Kang

**Affiliations:** 1Department of Pathology, School of Medicine, Case Western Reserve University, Cleveland, OH 44106, USAzxz972@case.edu (Z.Z.); yan.yan@mayo.edu (Y.Y.);; 2Byrd Alzheimer’s Center & Research Institute, Department of Molecular Medicine, USF Health Morsani College of Medicine, Tampa, FL 33613, USA; 3Louis Stokes Cleveland VA Medical Center, Cleveland, OH 44106, USA

**Keywords:** CHCHD10, TDP-43, PINK1, PARL, mitophagy

## Abstract

Coiled-coil-helix-coiled-coil-helix domain-containing 10 (CHCHD10) is a nuclear-encoded mitochondrial protein which is primarily mutated in the spectrum of familial and sporadic amyotrophic lateral sclerosis (ALS)–frontotemporal dementia (FTD). Endogenous CHCHD10 levels decline in the brains of ALS–FTD patients, and the CHCHD10^S59L^ mutation in *Drosophila* induces dominant toxicity together with PTEN-induced kinase 1 (PINK1), a protein critical for the induction of mitophagy. However, whether and how CHCHD10 variants regulate mitophagy flux in the mammalian brain is unknown. Here, we demonstrate through in vivo and in vitro models, as well as human FTD brain tissue, that ALS/FTD-linked CHCHD10 mutations (R15L and S59L) impair mitophagy flux and mitochondrial Parkin recruitment, whereas wild-type CHCHD10 (CHCHD10^WT^) normally enhances these measures. Specifically, we show that CHCHD10^R15L^ and CHCHD10^S59L^ mutations reduce PINK1 levels by increasing PARL activity, whereas CHCHD10^WT^ produces the opposite results through its stronger interaction with PARL, suppressing its activity. Importantly, we also demonstrate that FTD brains with TAR DNA-binding protein-43 (TDP-43) pathology demonstrate disruption of the PARL–PINK1 pathway and that experimentally impairing mitophagy promotes TDP-43 aggregation. Thus, we provide herein new insights into the regulation of mitophagy and TDP-43 aggregation in the mammalian brain through the CHCHD10–PARL–PINK1 pathway.

## 1. Introduction

One of the earliest abnormalities common to multiple neurodegenerative diseases, including frontotemporal dementia (FTD) and amyotrophic lateral sclerosis (ALS), is mitochondrial dysfunction, which is often detected by reduced mitochondrial respiration, calcium mishandling, lower membrane potential (Δψm), cristae structure disorganization, fission/fusion imbalance, and cytochrome c release (leading to apoptosis) [1,2,3]. Underlying these abnormalities are mitochondrial oxidative and proteotoxic stress [4] induced by the accumulation of misfolded proteins, such as ALS/FTD-associated TAR DNA-binding protein-43 (TDP-43) [5,6,7], which mislocalizes from the nucleus to the cytoplasm and associates with mitochondria, disrupting multiple facets of mitochondrial activity and transport [8,9,10].

The *CHCHD10* gene encoding a mitochondrial protein (coiled-coil-helix-coiled-coil-helix domain containing 10) is mutated in familial and sporadic FTD, ALS, and mixed FTD–ALS [11,12,13,14,15,16], linking CHCHD10 dysfunction to the etiology of these disorders. The estimated prevalence of *CHCHD10* mutations is 7.7% among FTD in a Chinese cohort [17] and 0.68–2.6% among FTD–ALS patients of European descent [11,12,13,14,15,16]. Many studies have implicated CHCHD10 in mitochondrial cristae morphology maintenance, mitochondrial dynamics, mitochondrial biogenesis, and respiratory chain complex activity [11,18,19,20,21,22]. Accordingly, ALS/FTD-linked CHCHD10 mutations impair mitochondrial respiratory activity [20,23], induce cristae disorganization [11,18], disrupt mitochondrial fusion [20], and promote mitochondrial fragmentation [22,23]. Furthermore, CHCHD10 variants physically and functionally interact with TDP-43. Specifically, FTD/ALS-linked CHCHD10^R15L^ and CHCHD10^S59L^ promote cytoplasmic and mitochondrial accumulation of misfolded TDP-43 [22,23], whereas wild-type CHCHD10 (CHCHD10^WT^) mitigates TDP-43 aggregation and TDP-43-induced mitochondrial dysfunction [20,22,23,24].

As first lines of defense in response to mitochondrial insults by oxidative and proteotoxic stress, the mitochondrial unfolded response (UPR^mt^), mitochondrial fusion, and the generation of mitochondrial-derived vesicle (MDV) pathways can repair mild insults to mitochondria [25,26,27]. However, excessive mitochondrial damage ultimately renders mitochondria unsalvageable, and the entire organelle together with its proteotoxic content must be degraded in bulk by the mitophagy machinery [28,29]. At the center of a major pathway are PTEN-induced putative kinase 1 (PINK1) and Parkin (PARK2) [30]. In healthy mitochondria, the 66 kDa full-length PINK1 is normally imported into the mitochondrial inner membrane (MIM), where it is cleaved within its transmembrane domain by the intramembrane protease presenilin-associated rhomboid-like protein (PARL) to generate an N-terminally truncated 55 kDa PINK1 fragment [31,32], which retro-translocates back to the cytosol and is rapidly degraded by the proteasome [33,34]. In damaged mitochondria, however, mitochondrial import of full-length PINK1 is suppressed while MIM-associated unprocessed PINK1 retro-translocates to the mitochondrial outer membrane (MOM), resulting in the accumulation of PINK1 on the MOM surface [32,34]. Upon activation of PINK1 by autophosphorylation, activated PINK1 recruits the E3 ligase Parkin to the MOM [35,36], which triggers the ubiquitination of MOM proteins and subsequent steps in mitophagy.

A recent study showed that the ALS/FTD-linked CHCHD10^S59L^ mutation exerts mitochondrial toxicity in part through PINK1 in *Drosophila* and Hela cells [32]. However, the mechanistic basis of CHCHD10^WT^ and ALS/FTD-linked CHCHD10 mutations in regulating the mitophagy process is largely unknown. Furthermore, the role of CHCHD10 variants in regulating mitophagy flux in the brain has not been investigated. CHCHD10 variants regulate MIC60 (also known as mitofilin) within the complex of the mitochondrial contact site and cristae organizing system (MICOS) in mitochondria, as we have reported previously [20]. MIC60 plays a crucial role in the recruitment of Parkin and the maintenance of PINK1 stability throughout mitophagy [37], indicating a potential association between CHCHD10 variants and mitophagy. Here, we crossed the mito-QC mitophagy flux reporter mice with transgenic mice neuronally expressing CHCHD10^WT^, CHCHD10^R15L^, or CHCHD10^S59L^ variants. We show that CHCHD10^WT^ promotes mitophagy flux in the brain, whereas ALS/FTD-linked CHCHD10^R15L^ and CHCHD10^S59L^ mutants strongly inhibit mitophagy flux. Mechanistically, CHCHD10^WT^ increases, whereas CHCHD10^R15L^ and CHCHD10^S59L^ mutants decrease, PINK1 levels and mitochondrial Parkin recruitment by differentially regulating PARL cleavage and activity. We also show that depletion of endogenous CHCHD10 increases PARL cleavage, reduces PINK1 levels, and impairs mitophagy flux, similar to ALS/FTD-linked mutations. Disrupting mitophagy through the knockdown of PINK1, CHCHD10, or MIC60 increases the accumulation of aggregated TDP-43. Thus, we demonstrate herein new insights into the regulation of mitophagy and TDP-43 aggregation through the CHCHD10–PARL–PINK1 pathway.

## 2. Methods

### 2.1. Ethics Approval

All methods and protocols used in this study involving mice were approved by the Institutional Animal Care and Use Committee (IACUC), and all methods were carried out following the relevant guidelines and regulations, which were also approved by the IACUC and Institutional Biosafety Committees (IBC).

### 2.2. Human Brain Samples

Frozen tissues from the frontal gyrus of FTLD-TDP and non-demented controls were provided by Drs. Allan Levey and Marla Gearing at Emory Alzheimer’s Disease Research Center (ADRC). In the procurement phase, the Emory ADRC utilized pathology-verified FTLD-tau and non-demented control tissues and attempted to match them as closely as possible for sex, age, and APOE genotypes (P50 AG025688).

### 2.3. Mice

Wild-type C57BL/6J, CHCHD10^WT^, CHCHD10^R15L^, and CHCHD10^S59L^ mice and mito-QC mice were bred in the C57BL/6J background. CHCHD10 expression profiles and pathology in transgenic mice expressing the CHCHD10 variants were previously described [20,24]. Mice were kept in sterile plastic cages with pelleted bedding in an SPF facility with 2–4 littermates until they were euthanized by transcardial perfusion or isoflurane overdose, along with cervical dislocation. There was unlimited access to food and water, a 12 h cycle of light and dark, and enrichment in the form of a mouse igloo. Based on age and genotype, available mouse littermates were divided into experimental groups for the relevant experiments. Direct mouse studies were carried out by people who were blind to their genotypes. All mice had been healthy up until the point of euthanasia and had no obvious abnormalities.

### 2.4. Antibodies and Reagents

Mouse Anti-Flag M2 (Cat#: F3165) and β-actin (Cat#: A2228) monoclonal antibodies were obtained from Sigma-Aldrich (St. Louis, MO, USA). Anti-TDP-43 (G400, Cat#: 3448), anti-PINK1 (Cat# 6946), anti-phospho-PINK1 (Ser228) (Cat# 46421), anti-GFP (Cat# 2555), and anti-Parkin 9 (Cat# 2132) antibodies were purchased from Cell Signaling (Danvers, MA, USA). Anti-human TDP-43 antibody (Cat#: H00023435-M01) was purchased from Abnova (Walnut, CA, USA). Anti-CHCHD10 antibody (Cat#: ab121196) was purchased from Abcam (Cambridge, UK). Anti-Tom20 (Cat#: sc-17764), and MIC60 (Cat#: sc-390706) antibodies were purchased from Santa Cruz Biotechnology (Dallas, TX, USA). Anti-PARL (Cat# 26679-1-AP) antibody was purchased from Proteintech (Rosemont, IL, USA).

### 2.5. Cell Culture

HEK293T and HT22 cells were cultured in Dulbecco’s modified Eagle’s medium (DMEM, Thermo Scientific, Waltham, MA, USA) supplemented with 10% fetal bovine serum (FBS) and 1% penicillin/streptomycin (P/S). All cells were cultured in a humidified atmosphere (5% CO_2_) at 37 °C.

### 2.6. DNA Constructs, Transfections, and rAAV9 Transduction

P3X-Flag-CHCHD10^WT^, P3X-Flag-CHCHD10^R15L^, and P3X-Flag-CHCHD10^S59L^ constructs have previously been described [20,22,24]. For DNA plasmid transfections, HEK293T and HT22 cells were transfected with Fugene HD (Promega, Madison, WI, USA) in Opti-MEM I (Invitrogen, Carlsbad, CA, USA) according to the manufacturer’s instructions and harvested 48 h post-transfection. For siRNA transfections, lipofectamine 2000 (Invitrogen, Carlsbad, CA, USA) was used according to the manufacturer’s instructions.

### 2.7. Mitochondria Isolation, Cell/Tissue Lysis, and SDS-PAGE

Mitochondria isolation was performed following the manufacturer’s instructions using mitochondria isolation kits for cultured cells and for tissue (Thermo Scientific, Waltham, MA, USA). Cells and brain tissues were lysed with RIPA lysis buffer (50 mM Tris pH 7.4, 150 mM NaCl, 2 mM EDTA, 1% NP-40, 0.1% sodium dodecyl sulfate) [20,24]. The concentrations of total protein were quantified using a colorimetric detection assay (BCA Protein Assay, Pierce, Waltham, MA, USA). Protein lysates in equal amounts were separated using sodium dodecyl sulfate (SDS)–polyacrylamide gel electrophoresis (PAGE) and transferred to a nitrocellulose membrane (Millipore Corporation, Bedford, MA, USA). For filter-trap assays (FTA), equal amounts of RIPA-soluble or sonicated RIPA-insoluble extracts were filtered through 0.2 mm cellulose acetate membranes [20,24] (ThermoFisher Scientific, Waltham, MA, USA) using a 96-well vacuum dot blot apparatus (Bio-Rad, Hercules, CA, USA), followed by PBS washing and 20% methanol fixation. Interested proteins were probed with primary antibodies, followed by peroxidase-conjugated secondary antibodies and ECL detection (Merck Millipore Corporation, Darmstadt, Germany). All immunoblot images were acquired using LAS-4000 (GE Healthcare Biosciences, Pittsburgh, PA, USA) or ImageQuant 800 (Amersham, Chicago, IL, USA) and quantified using ImageJ (NIH, Bethesda, MD, USA).

### 2.8. Immunocytochemistry, Proximity Ligation Assay, and Fluorescent Microscopy

For immunocytochemistry (ICC), cells were washed with PBS and fixed for 15 min at room temperature with 4% paraformaldehyde (PFA) [24]. Fixed cells were washed with PBS, incubated with blocking solution (0.2% Triton X-100, 3% normal goat serum) for 1 h, incubated with primary antibody at 4 °C overnight, washed three times with PBS, and incubated with Alexa-488 or Alexa-594-conjugated secondary IgG antibodies for 1 h at room temperature (Vector Laboratories, Burlingame, CA, USA). The slides were then washed three times with PBS before being mounted with a fluorochrome mounting solution (Vector Laboratories). Proximity ligation assays (PLA) were performed as previously described [38,39,40]. Briefly, cells were washed with PBS and fixed with 4% paraformaldehyde and blocked with 3% NGS. Indicated primary antibodies were applied followed by the Duolink^®^ In Situ PLA^®^ secondary antibody probes (Sigma-Aldrich) and DAPI. All images were captured with Nikon AX Ti2 confocal (Tokyo, Japan) or ZEISS LSM880 confocal microscopes (Oberkochen, Germany), and ImageJ software Version 1.54g 18 October 2023 (NIH, Bethesda, MD, USA) was used to quantify the immunoreactivities. In all ICC experiments, all comparison images had the same laser intensity, exposure time, and filter. During image acquisition and quantification, investigators were blinded to the experimental conditions, and regions of interest were selected randomly. Brightness/contrast adjustments were applied uniformly to all comparison images. The nucleus signals were not taken into account in the quantification of the images in the PLA experiments. Image J’s JACoP plugin was used to calculate the Manders overlap coefficient to quantify colocalizations (MOC) [24]. Identical thresholds were applied to each channel.

### 2.9. Statistical Analysis

All graphs were created and analyzed with GraphPad Prism 8.0 software (GraphPad Software, San Diego, CA, USA) utilizing the student’s *t*-test, one-way analysis of variance (ANOVA) followed by Dunnett or Sidak post hoc tests, or two-way ANOVA. When *p* < 0.05, differences were considered significant. All graphs were depicted as mean SEMs (error bars).

## 3. Results

### 3.1. CHCHD10^WT^ Promotes and ALS/FTD-Linked CHCHD10 Mutations Suppress Mitophagy Flux and Mitochondrial Parkin Recruitment In Vivo

CHCHD10 is a component of the mitochondrial contact site and cristae organizing system (MICOS) complex [11], which stabilizes the MICOS complex through its core protein MIC60, whereas ALS/FTD-linked CHCHD10 mutations (R15L and S59L) destabilize the MIC60 complex [20]. Previous studies have shown that MIC60 also regulates mitophagy through the PINK1–Parkin pathway [37,41]. Based on these observations, we wondered whether CHCHD10^WT^ and ALS/FTD-linked CHCHD10 mutations (R15L and S59L) impact mitophagy in the brain. To address this question, we took advantage of our transgenic mice neuronally expressing CHCHD10 variants (WT, R15L, and S59L) under the control of the mouse *PrP* promoter. These mice express CHCHD10 transcripts 2.5–2.8 times more than the endogenous CHCHD10 transcript, and the transgenic CHCHD10 proteins are expressed throughout the brain and spinal cord [20,24]. To determine the role of CHCHD10 in mitophagy flux in vivo, we crossed CHCHD10 transgenic mouse variants (WT, R15L, and S59L) with mito-QC homozygous mice expressing the pH-sensitive mCherry-GFP protein fused to the FIS1 mito-targeting sequence under the control of the ubiquitously expressed CAG promoter (Figure 1a) [42]. In mito-QC mice, the mitochondrial network fluoresces red and green (yellow) when mitophagy is inactive. During mitophagy, however, mitochondria are delivered to lysosomes, where mCherry fluorescence remains stable, but GFP fluorescence is quenched by the lysosomal acidic environment (Figure 1a) [42]. The numbers of acidified red-only (mCherry) puncta per field, therefore, represent a quantitative measure of mitophagy flux [42,43].

At 10 months of age, the cortex of CHCHD10^WT^ mice exhibited a significant 34% increase in red-only puncta compared to that of WT littermates (Figure 1b,c), indicating that CHCHD10^WT^ augments mitophagy flux. In contrast, CHCHD10^R15L^ and CHCHD10^S59L^ brains exhibited > 2.3-fold reductions in red-only puncta compared to CHCHD10^WT^ brains (Figure 1b,c), indicating that these ALS/FTD-linked CHCHD10 mutations disrupt mitophagy flux in the brain. Parkin recruitment to mitochondria is a key step in mitophagy initiation in the PINK1–Parkin pathway [30]. To gain insights into the mechanism underlying the changes in mitophagy flux, we isolated mitochondrial and cytosolic fractions from cortical tissues. Western blotting for Parkin showed that CHCHD10^WT^ significantly increases the ratio of mitochondrial to cytosolic Parkin (Figure 1d,e), whereas CHCHD10^R15L^ and CHCHD10^S59L^ mutations significantly reduce this ratio compared to CHCHD10^WT^ (Figure 1d,e). We also confirmed these in vivo findings in HEK293T (Figure 1f,g) and HT22 mouse neuroblastoma cells (Figure 1h,i) transfected with GFP-Parkin + CHCHD10 variants and treated with the mitochondrial uncoupler CCCP, where similar results were observed in both Western blotting (Figure 1f,g) and immunocytochemistry (ICC) (Figure 1h,i), respectively. The purity of mitochondrial and cytosolic fractions was also confirmed in vivo and in vitro (Appendix A). Thus, these results indicate that ALS/FTD-linked CHCHD10 mutations (R15L and S59L) impair mitophagy flux in the brain by reducing mitochondrial Parkin recruitment, whereas CHCHD10^WT^ enhances mitophagy flux by increasing Parkin recruitment.

### 3.2. ALS/FTD-Linked CHCHD10 Mutations (R15L and S59L) Increase PARL Activity and Reduce Activated Full-Length PINK1 In Vivo

PINK1 stability and its ability to recruit Parkin to mitochondria are controlled by the mitochondrial inner membrane (MIM) protease PARL, which, by cleaving the N-terminus of PINK1, allows the retro-translocation of PINK1 to the cytosol [31,32] to promote its rapid degradation by the proteasome [33,34]. PARL undergoes rapid processing to its mature 38 kDa form (called PARL-MAMP), which then can be autocatalytically further cleaved [44], generating the 33 kDa fragment (PARL-PACT) [45,46]. To gain further insight into the mechanism underlying the changes in mitochondrial Parkin recruitment and mitophagy flux by CHCHD10 variants, we immunoblotted WT, CHCHD10^WT^, CHCHD10^R15L^, and CHCHD10^S59L^ mouse brain cortical lysates for PARL. The 38kDa PARL-MAMP was significantly elevated by approximately two-fold in CHCHD10^WT^ brains compared to WT brains, whereas CHCHD10^S59L^ brains exhibited significantly less PARL-MAMP compared to CHCHD10^WT^ brains (Figure 2a,b). CHCHD10^R15L^ brains showed less PARL-MAMP than CHCHD10^WT^ brains but without reaching statistical significance (Figure 2a,b). In contrast, both CHCHD10^R15L^ and CHCHD10^S59L^ brains exhibited approximately three-fold increases in the 33 kDa PARL-PACT compared to CHCHD10^WT^ brains, whereas CHCHD10^WT^ brains showed no significant change in this fragment compared to WT brains (Figure 2a,b). These results therefore collectively suggest that CHCHD10 variants differentially modulate PARL cleavage and activity.

We next immunoblotted the same mouse brain cortical lysates for pS228-phosphorylated PINK1 (pS228-PINK1), representing the autocatalytically activated form of PINK1 which is capable of binding Parkin [35]. Both full-length (fl) 66 kDa and cleaved (cl) 55 kDa forms of pS228-PINK1 were readily detected (Figure 2a). Specifically, the 66kDa fl-pS228-PINK1, which associates with the MOM and recruits Parkin to mitochondria [35,47], was significantly reduced in CHCHD10^R15L^ and CHCHD10^S59L^ but not CHCHD10^WT^ brains compared to WT brains (Figure 2a,c). In contrast, the 55 kDa cl-pS228-PINK1, which does not associate with MOM and represses mitochondrial Parkin recruitment [35,47], was disproportionately reduced in CHCHD10^WT^ brains compared to WT brains (Figure 2a,c). This phosphorylated 55 kDa cl-PINK1 fragment was further reduced in CHCHD10^R15L^ and CHCHD10S^59L^ brains but generally in proportion to the reduction in 66 kDa fl-pS228-PINK1 (Figure 2a,c). A band of unknown significance slightly smaller than the 66 kDa fl-pS228-PINK1 was also seen (Figure 2a). While the total 66kDa fl-PINK1 levels generally corresponded to 66 kDa fl-pS228-PINK1 levels across genotypes (Figure 2a; Appendix A), total cleaved 55 kDa cl-PINK1 was not detected through Western blotting (Figure 2a), likely owing to its rapid proteolysis [33,34]. These results suggest that through modulating PARL cleavage and activity, CHCHD10^R15L^ and CHCHD10^S59L^ mutations but not CHCHD10^WT^ reduce the 66 kDa active form of fl-PINK1, whereas CHCHD10^WT^ preferentially reduces the inhibitory 55 kDa form of cl-PINK1.

### 3.3. CHCHD10 Interactions with PARL and PINK1 Are Suppressed by CHCHD10^R15L^ and CHCHD10^S59L^ Mutations

As CHCHD10 variants differentially altered PARL cleavage and PINK1 levels, we wondered if CHCHD10 forms a complex with PARL. To test this possibility, we used the proximity ligation assay (PLA) to detect PARL–CHCHD10 complexes in cells transfected with Flag-CHCHD10 variants and treated with and without CCCP. Without CCCP treatment, we detected weak PARL + Flag-CHCHD10 (M2 antibody) PLA signals across all CHCHD10 variants, although PARL complexes with CHCHD10^R15L^ and CHCHD10^S59L^ were significantly weaker than those with CHCHD10^WT^ (Figure 3a,b). Upon CCCP treatment, PARL complexes with all Flag-CHCHD10 variants increased; however, the magnitude and significance for PARL’s preferential interaction with CHCHD10^WT^ further increased compared to CHCHD10^R15L^ and CHCHD10^S59L^ (Figure 3a,b). Negative controls of PARL + M2 Flag-antibody PLA in vector control-transfected cells showed no detectable PLA signal, indicating the specificity of PLA (Figure 3a) We also carried out co-immunoprecipitation (co-IP) experiments under CCCP treatment to confirm the interactions seen in PLA. Again, we observed markedly more PARL pulldown in M2 Flag-antibody immune complexes of CHCHD10^WT^-transfected cells compared to those transfected with ALS/FTD-linked mutants (R15L and S59L) (Appendix A), indicating that CHCHD10^WT^ forms a more stable complex with PARL than CHCHD10^R15L^ and CHCHD10^S59L^. As PARL transiently associates with PINK1 to mediate its cleavage [34], PLA experiments after CCCP treatment showed that PINK1 also preferentially associates with CHCHD10^WT^ compared to CHCHD10^R15L^ and CHCHD10^S59L^ (Figure 3c,d), similar to PARL. These results suggest that CHCHD10 variants differentially regulate the cleavage and activity of PARL through their physical association, with decreased interaction corresponding to greater PARL cleavage and reduced activated fl-PINK1 in the brain.

### 3.4. CHCHD10 Variants Depend on PARL to Regulate Mitochondrial Parkin Recruitment

Next, we tested whether the regulation of Parkin recruitment by Flag-CHCHD10 variants is dependent on PARL activity. To test this, HT22 cells were co-transfected with GFP-Parkin and CHCHD10 variants with or without PARL siRNA for 48 h and treated with CCCP. In control siRNA-transfected cells, CHCHD10^WT^, but not CHCHD10^R15L^ or CHCHD10^S59L^, increased mitochondrial Parkin recruitment, as detected by GFP-Parkin colocalization with the mitochondrial marker Tom20 (Figure 4a,b). In PARL siRNA-transfected cells, however, mitochondrial recruitment of GFP-Parkin increased overall as previously reported [34], and we found no significant differences in GFP-Parkin colocalization with Tom20 between the CHCHD10 variants (Figure 4a,b), indicating that CHCHD10-regulated Parkin recruitment is dependent on PARL activity. As expected, PARL siRNA effectively decreased both PARL-MAMP and PARL-PACT levels (Figure 4c). These results indicate that CHCHD10 variants differentially regulate Parkin recruitment through PARL.

### 3.5. CHCHD10 Depletion Enhances PARL Cleavage and Reduces PINK1, Suppressing Parkin Recruitment and Mitophagy Flux

While CHCHD10^R15L^ and CHCHD10^S59L^ are rare mutations found in familial ALS–FTD patients, endogenous CHCHD10 is significantly downregulated in brains of TDP-43 transgenic mice and sporadic ALS and FTD patients who do not carry CHCHD10 mutations [20,24,48]. To determine whether endogenous CHCHD10 alters PARL and PINK1 levels, we knocked down CHCHD10 using siRNA in HEK293T cells, treated cells with and without CCCP, and assessed PARL and PINK1 levels through immunoblotting. In HEK293T cells, the 38 kDa PARL-MAMP was not detected (Figure 5a). Nevertheless, knockdown of CHCHD10 significantly increased the 33 kDa PARL-PACT and significantly reduced fl-PINK1 levels both in the presence or absence of CCCP (Figure 5a–c). The rapidly degraded cl-PINK1 was barely detectable under all conditions (Figure 5a). This indicated that endogenous CHCHD10 depletion inhibits PARL activity to control PINK1 levels, as in CHCHD10^R15L^ and CHCHD10^S59L^ mutations. As CHCHD10 depletion reduced PINK1 levels, mitochondrial recruitment of GFP-Parkin was also significantly reduced, as evidenced by an >50% reduction in the ratio of mitochondrial to cytosolic GFP-Parkin by CHCHD10 siRNA compared to control siRNA (Figure 5d,e). To determine if endogenous CHCHD10 depletion likewise retards mitophagy flux, we utilized the mtKeima probe, a mitochondria-targeted pH-sensitive dual excitation reporter construct that enables the tracking of mitophagy flux in live cells. The mtKeima protein is excited at 443 nm at neutral pH (mitochondria), whereas it is excited at 580 nm in acidic pHs (lysosome), and both excitations maximally emit at 620 nm [49,50]. Hence, the ratio of 580 nm to 443 nm excitations provides a measure of mitophagy flux. Indeed, the ratio of 580 nm to 443 nm (pseudo-colored to red and green, respectively) steadily increased from 0 to 30 min after CCCP treatment in control shRNA lentivirus transduced cells, whereas this ratio did not increase with the same CCCP treatment in CHCHD10 shRNA lentivirus transduced cells (Figure 5f,g), indicating that CHCHD10 depletion impairs mitophagy flux. These results taken together indicate that endogenous CHCHD10 promotes Parkin recruitment and mitophagy flux through the PARL–PINK1 pathway.

### 3.6. Disruption of the PARL–PINK1 Pathway in FTLD-TDP Patient Brains

We and others have previously shown that ALS and frontotemporal lobar degeneration associated with TDP-43 (FTLD-TDP) patient brains exhibit significant reductions in CHCHD10 levels [20,24,48]. However, the role of PARL and PINK1 in FTLD-TDP is unknown. Thus, we examined the level and cleavage pattern of PARL and PINK1 in the frontal gyrus of 10 FTLD-TDP and 11 control patients (Appendix A, case information). In these brains, we observed little to no detectable 38 kDa PARL-MAMP (Figure 6a). However, the 33 kDa PARL-PACT was significantly elevated by >1.7-fold in FTLD-TDP brains compared to control brains (Figure 6a,b). In contrast, the 66 kDa fl-PINK1 was significantly reduced by 28% in FTLD-TDP versus controls (Figure 6a,c). Surprisingly, FTLD-TDP brains exhibited a nearly four-fold increase in the ~55 kDa cl-PINK1 compared to controls (Figure 6a,d), likely reflecting both increased PARL-mediated PINK1 cleavage and impaired proteasome function that rapidly degrades cl-PINK1 [33,34]. Together with previous observations of reduced CHCHD10 in FTLD-TDP brains [20], these results indicate that the CHCHD10-regulated PARL–PINK1 pathway is disrupted in FTLD-TDP brains.

### 3.7. Depletion of PINK1, CHCHD10, or MIC60 Promotes TDP-43 Aggregation

A hallmark of TDP-43 pathology is the cytoplasmic mislocalization and aggregation of TDP-43, a bulk of which is associated with mitochondria [22,51]. TDP-43 is proteolytically processed in mitochondria, and it interacts with multiple mitochondrial proteins [52], including CHCHD10 [20,22]. Thus, we wondered whether mitophagy mediated by PINK1, CHCHD10, or MIC60 alters the aggregation of TDP-43. MIC60, a MIM protein constituting the core structural component of the MICOS complex, stabilizes PINK1 and promotes mitochondrial Parkin recruitment [37]. To test whether mitophagy disruption through these proteins alters TDP-43 aggregation, we co-transfected TDP-43 with control siRNA or siRNAs for PINK1, CHCHD10, or MIC60 in HEK293T cells, after which mitophagy was induced through CCCP treatment for 4 h. From RIPA-soluble and RIPA-insoluble lysates, we then detected TDP-43 aggregates via filter-trap assays and measured protein expression using Western blotting. PINK1 knockdown led to significant 1.5- and 2.2-fold increases in the capture of TDP-43 aggregates from RIPA-soluble and RIPA-insoluble fractions, respectively, in filter trap assays (Figure 7a,b). While the increase in TDP-43 aggregates due to CHCHD10 knockdown in the soluble fraction did not reach statistical significance (Figure 7a,c), CHCHD10 knockdown increased TDP-43 aggregates by a significant four-fold in the insoluble fraction (Figure 7a,c). Similar to PINK1 knockdown, MIC60 knockdown resulted in significant 2.2- and 1.5-fold increases in TDP-43 aggregate capture from the soluble and insoluble fractions, respectively (Figure 7a,d). We also confirmed the knockdown of corresponding proteins through Western blotting (Appendix A). Notably, TDP-43 levels detected in the soluble fraction using Western blotting were lower after CHCHD10 or MIC60 siRNA (Appendix A), despite the observed increases in TDP-43 aggregates due to their knockdown (Figure 7a,c,d). These results therefore indicate that endogenous PINK1, CHCHD10, and MIC60 inhibit TDP-43 aggregation at least in part by promoting mitophagy.

## 4. Discussion

The critical importance of mitophagy through the PINK1–Parkin pathway is widely exemplified by loss-of-function mutations in PINK1 and Parkin in Parkinson’s disease (PD) [53]. However, little is known about the role of mitophagy in the spectrum of ALS–FTD [54]. CHCHD10, a mitochondrial protein mutated in the ALS–FTD spectrum, has been shown to impact multiple facets of mitochondrial function. Specifically, as a component of the MIC60-based MICOS complex [11,18], CHCHD10 stabilizes the MICOS complex and the MIC60–Opa1 complex [20]. MIC60 also regulates mitophagy through the PINK1–Parkin pathway [37,41], and ALS/FTD-linked CHCHD10^S59L^ mutation induces mitochondrial toxicity in part through PINK1 (32). In this study, we set out to investigate the role of CHCHD10^WT^ and ALS/FTD-linked CHCHD10 mutations in mitophagy. We made a series of discoveries that further our mechanistic understanding of CHCHD10 in mitophagy regulation and how mitophagy could impact pathology in the ALS–FTD spectrum. By utilizing the mito-QC reporter mouse crossed with transgenic models neuronally expressing CHCHD10 variants, we showed that ALS/FTD-linked CHCHD10 mutations (R15L and S59L) impede mitophagy flux, whereas CHCHD10^WT^ enhances it. We uncovered the mechanistic basis of these phenotypes by demonstrating that CHCHD10^R15L^ and CHCHD10^S59L^ mutations or depletion of endogenous CHCHD10 increase PARL-PACT and reduce activated fl-PINK1, resulting in impaired mitochondrial Parkin recruitment and mitophagy flux. Compared to CHCHD10^R15L^ and CHCHD10^S59L^ mutations, CHCHD10^WT^ binds more strongly to PARL and suppresses its activity, which is associated with increased Parkin recruitment and mitophagy flux. Conversely, endogenous CHCHD10 depletion increases PARL-PACT, and reduces fl-PINK1, hindering Parkin recruitment and mitophagy flux. We also showed for the first time that FTLD-TDP brains exhibit elevated levels of PARL-PACT, reduced levels of fl-PINK1, and highly elevated levels of cl-PINK1, providing new insights into the deregulation of the PARL–PINK1 pathway in the FTLD-TDP disease process. Finally, we showed that disrupting mitophagy through depletion of PINK1, CHCHD10, or MIC60 all result in the accumulation of aggregated TDP-43, the pathological hallmark of ALS and FTLD-TDP, demonstrating that mitophagy-related defects impact the disease process.

CHCHD10 mutations (R15L and S59L) enhanced PARL cleavage (PARL-PACT) while exhibiting reduced PARL binding compared to CHCHD10^WT^. Endogenous CHCHD10 depletion also enhanced PARL-PACT levels, similar to CHCHD10 mutations. These results suggest that endogenous CHCHD10 normally suppresses PARL cleavage and that the loss and/or aberrant binding of CHCHD10 mutants to PARL enhances its cleavage. Increased levels of 33 kDa PARL-PACT due to CHCHD10 mutations also associated with decreased full-length activated pS228-PINK1 and total full-length PINK1, which suggests that the 33 kDa PARL-PACT may be more active toward PINK1 cleavage than its 38 kDa precursor (PARL-MAMP). This notion is supported by previous studies showing that PARL-PACT is generated by autoproteolysis [44] and that PARL-PACT is more active than its precursor (PARL-MAMP) in mitochondrial fragmentation [46] and toward PINK1 cleavage in kinetic proteolytic assays using recombinant PARL proteins [45]. Alternatively, the differential binding of CHCHD10 variants to PINK1 per se may also contribute to the differences in PINK1 activation and its cleavage by PARL. While the 55kDa-cleaved PINK1 protein was not detected in the brain, both 66kDa and 55kDa forms of activated pS228-PINK1 were seen, and CHCHD10^WT^ brains showed disproportionately decreased levels of the 55 kDa cl-pS228-PINK1 fragment compared to WT brains. As the 55 kDa cl-PINK1 fragment is unable to associate with MOM but binds Parkin and hinders Parkin recruitment to mitochondria [47], reduced levels of 55 kDa pS228-cl-PINK1 fragment in CHCHD10^WT^ brains may contribute to enhanced mitochondrial Parkin recruitment and mitophagy flux. In contrast, CHCHD10 mutations (R15L and S59L) appear to reduce the 66 kDa activated fl-pS228-PINK1 to impair Parkin recruitment and mitophagy flux.

A recent study by Baek and colleagues showed that CHCHD10^S59L^ overexpression in HeLa cells stably expressing YFP-Parkin promotes constitutive mitochondrial YFP-Parkin and LC3 recruitment in the absence of mitophagy induction (i.e., CCCP). However, CHCHD10^S59L^ overexpression also promoted LC3 accumulation and its mitochondrial recruitment even in Parkin-deficient HeLa cells [23]. This indicates that the mode of mitophagy induction in HeLa cells may be distinct from the depolarization-induced PINK1–Parkin pathway. It is also notable that unlike HEK293T and HT22 cells used in this study, Hela cells do not exhibit PARL-MAMP cleavage to generate PARL-PACT [44], a critical difference that may alter the effects of CHCHD10^S59L^ on mitophagy in a cell-type dependent manner. In the current study, we modeled mitophagy induced by mitochondrial depolarization (CCCP) in cultured cells and chronic mitochondrial proteotoxic stress in vivo. Indeed, 10-month-old CHCHD10^R15L^ and CHCHD10^S59L^ mice accumulate substantial pathology, and both CHCHD10 and TDP-43 aggregates associated with mitochondria and exhibited electrophysiological and behavioral deficits that were pathologically pertinent to ALS–FTD [24]. The study by Baek et al. also showed that ectopic expression of *Drosophila CG5010* (an ortholog of mammalian *CHCHD10* and *CHCHD2*) carrying the S81L mutation (orthologous to human S59L mutation) induces toxicity in *Drosophila* that is rescued through the knockdown of PINK1 [23]. Although they did not directly study mitophagy in *Drosophila*, it is unclear if the *CG5010* S81L mutation phenocopies the orthologous mutation in mammalian CHCHD10 or CHCHD2, since *Drosophila* carries only one ortholog of both mammalian genes. Alternatively, as *Drosophila* PINK1 appears to promote mitochondrial fission while mammalian PINK1 promotes fusion [55], *Drosophila* PINK1 and mutant *CG5010* may functionally interact in ways distinct from those in mammalian systems in vivo. Hence, the apparent discrepancies between this study and the study by Baek and colleagues likely reflect differences in basal versus induced states of mitophagy, cell types used, and species examined.

Despite the numerous studies of the PINK1–Parkin pathway in PD [53], the role of PARL and PINK1 in ALS or FTLD-TDP is essentially unknown. In this study, we showed for the first time that the 33 kDa PARL-PACT and the 55 kDa cl-PINK1 are increased, while the 66 kDa fl-PINK1 is decreased in FTLD-TDP brains. This finding is consistent with the reduction in CHCHD10 levels in FTLD-TDP brains [20,24] and the current observation that CHCHD10 depletion increases PARL-PACT and reduces fl-PINK1, hindering Parkin recruitment and mitophagy flux. FTLD-TDP brains accumulated copious amounts of the 55 kDa cl-PINK1, which otherwise should be rapidly degraded [33,34]. This indicates that FTLD-TDP brains are unable to effectively remove this cl-PINK1 fragment through the proteasome, perhaps due to the loss of proteasome activity, a finding consistent with the accumulation of cytosolic cl-PINK1 due to TDP-43 overexpression [56]. Thus, the depletion of CHCHD10 [20,24] and fl-PINK1 together with the accumulation of cl-PINK1, the latter of which actively represses mitochondrial Parkin recruitment [47], collectively indicate that FTLD-TDP brains are likely deficient in mitophagy. In this vein, our observation that disrupting mitophagy through depletion of PINK1, CHCHD10, or MIC60 induces the accumulation of aggregated TDP-43 takes on added significance, suggesting that deficient mitophagy through the CHCHD10–PARL–PINK1 pathway plays a significant role in TDP-43 pathogenesis in FTLD-TDP. Moreover, our previous observation that CHCHD10^WT^ overexpression in the brain protects against TDP-43 pathology [24] is consistent with the findings in this study, in which both CHCHD10^WT^ overexpression and endogenous CHCHD10 promote mitophagy. Thus, therapeutic strategies to boost CHCHD10 and/or PINK1 levels or activity could provide simultaneous protection against the accumulation of TDP-43 and defective clearance of dysfunctional mitochondria in ALS-FTD.

## 5. Conclusions

In summary, our findings indicate that CHCHD10 ALS/FTD-associated CHCHD10 mutations (R15L and S59L) and CHCHD10 depletion inhibit Parkin recruitment during mitophagy by increasing PARL cleavage and decreasing PINK1 levels. This leads to a disruption in mitophagy and the aggregation of TDP-43. On the contrary, when CHCHD10^WT^ is overexpressed, it enhances mitophagy and Parkin recruitment by inhibiting PARL cleavage and increasing PINK1 stability. Hence, increasing CHCHD10 may offer a promising strategy for combating TDP-43 aggregation and mitophagy impairment in ALS–FTD.

## Figures and Tables

**Figure 1 cells-12-02781-f001:**
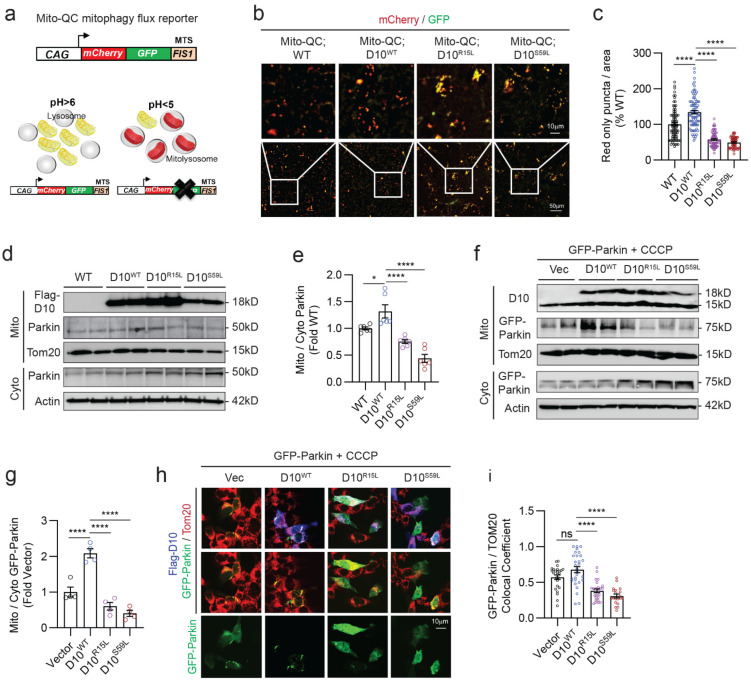
CHCHD10^WT^ promotes and ALS/FTD-linked CHCHD10 mutations suppress mitophagy flux and mitochondrial Parkin recruitment in vivo. (**a**) Schematic representation of the mito-QC mitophagy flux reporter principle. (**b**) Representative mCherry (red) and GFP (green) images of cortical brain sections from 10-month-old mito-QC mice or mito-QC mice expressing CHCHD10 variants (WT, R15L, or S59L). Bottom panel boxed areas are magnified in upper panels. (**c**) Quantification of red only (mCherry) puncta per area (one-way ANOVA, F(3, 303) = 98.18, *p* < 0.0001; posthoc Dunnett, **** *p* < 0.0001, *n* = 58–85 images/genotype from 4 mice/genotype). (**d**) Mitochondrial and cytosolic fractions isolated from 10-month-old WT and CHCHD10 Tg mice (WT, R15L, and S59L) and immunoblotted for CHCHD10, Parkin, Tom20, and Actin. (**e**) Quantification of the mitochondrial to cytosolic Parkin ratio (one-way ANOVA, F(3, 20) = 24.99, *p* < 0.0001; posthoc Dunnett, * *p* < 0.05, **** *p* < 0.0001, *n* = 6 mice/genotype). (**f**) HEK293T cells co-transfected with GFP-Parkin and vector control or Flag-CHCHD10 variants (WT, R15L, and S59L), treated with CCCP (10 mM, 4 h), subjected to isolation of mitochondrial and cytosolic fractions, and immunoblotted for the indicated proteins. (**g**) Quantification of the mitochondrial to cytosol Parkin ratio (one-way ANOVA, F(3, 12) = 40.62, *p* < 0.0001; posthoc Dunnett, *****p* < 0.0001, *n* = 4 replicates). (**h**) Representative images of HT22 cells transfected with GFP-Parkin (green) and vector control or Flag-CHCHD10 variants (WT, R15L, or S59L), treated with CCCP (10 mM, 4 h), and immunostained for Flag-CHCHD10 (blue) and Tom20 (red). (**i**) Quantification of GFP-Parkin colocalization with Tom20 (one-way ANOVA, F(3, 95) = 23.76, *p* < 0.0001; posthoc Dunnett, **** *p* < 0.0001, ns = not significant, *n* = 20–29 images/condition from 3 experiments). Error bars = SEM.

**Figure 2 cells-12-02781-f002:**
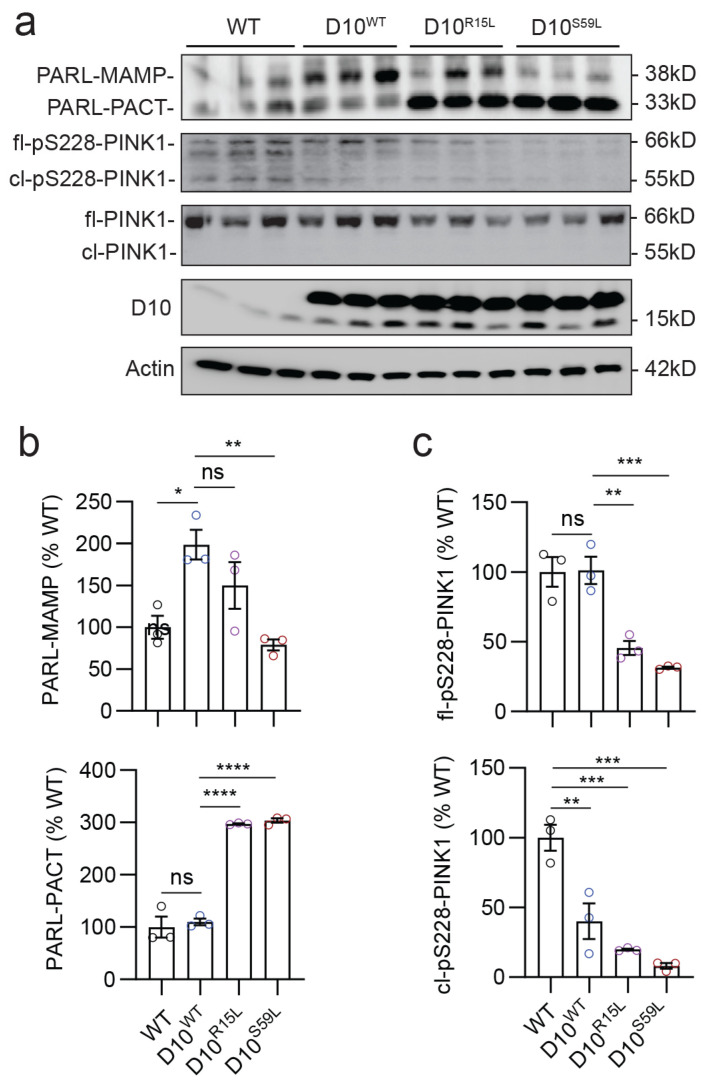
CHCHD10^R15L^ and CHCHD10^S59L^ mutations increase PARL cleavage and reduce activated PINK1 in vivo. (**a**) Representative blots of RIPA-soluble lysates from the cortex of 10-month-old WT and CHCHD10 Tg mice (WT, R15L, and S59L) immunoblotted for PARL, pS228-PINK1, PINK1, CHCHD10, and Actin. (**b**) Quantification of 38 kDa PARL-MAMP and 33 kDa cleaved PARL-PACT (PARL-MAMP: one-way ANOVA, F(3, 8) = 8.722, *p* = 0.0067; PARL-PACT: F(3, 8) = 113.6, *p* < 0.0001, *n* = 3 mice/genotype). (**c**) Quantification of full-length S228-phosphorylated PINK1 (fl-pS228-PINK1) and cleaved S228-phosphorylated PINK1 (cl-pS228-PINK1) (fl-pS228-PINK1 one-way ANOVA, F(3, 8) = 22.48, *p* = 0.0003; cl-pS228-PINK1: F(3, 8) = 56.24, *p* < 0.0001; Posthoc Dunnett, * *p* < 0.05, ** *p* < 0.01, *** *p* < 0.001, **** *p* < 0.0001, ns = not significant, *n* = 3 mice/genotype).

**Figure 3 cells-12-02781-f003:**
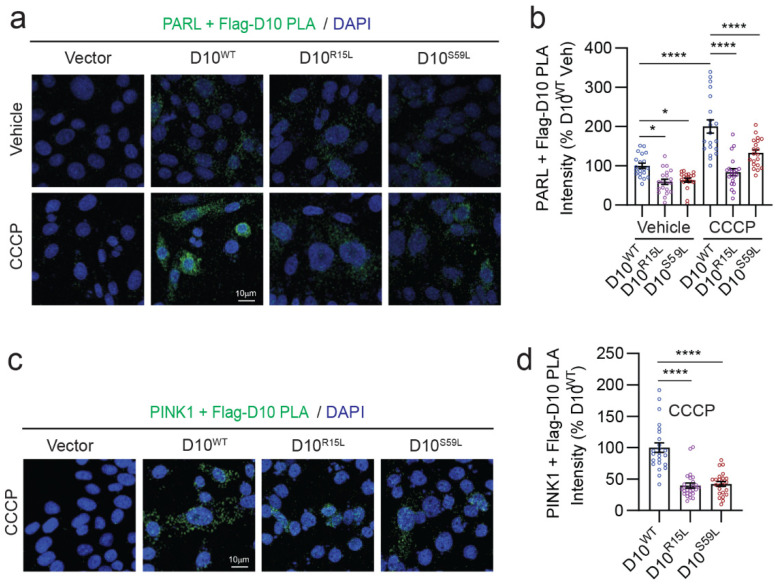
CHCHD10 interactions with PARL and PINK1 are suppressed by CHCHD10^R15L^ and CHCHD10^S59L^ mutations. (**a**) Representative images of proximity ligation assays (PLA) showing interactions between Flag-CHCHD10 (M2 Flag) and PARL in HT22 cells transfected with Flag-CHCHD10 variants + CCCP treatment (1 mM, 12 h). (**b**) Quantification of PLA intensity per image (one-way ANOVA, F(5, 114) = 31.49, *p* < 0.0001; posthoc Dunnett, * *p* < 0.05, **** *p* < 0.0001, *n* = 20 images from four replicates). (**c**) Representative images of PLA showing interactions between Flag-CHCHD10 (M2 Flag) and PINK1 in HT22 cells transfected with Flag-CHCHD10 variants + CCCP treatment (1 mM, 12 h). (**d**) Quantification of PLA intensity per image (one-way ANOVA, F(2, 72) = 40.82, *p* < 0.0001; posthoc Dunnett, **** *p* < 0.0001, *n* = 24–27 images from four replicates).

**Figure 4 cells-12-02781-f004:**
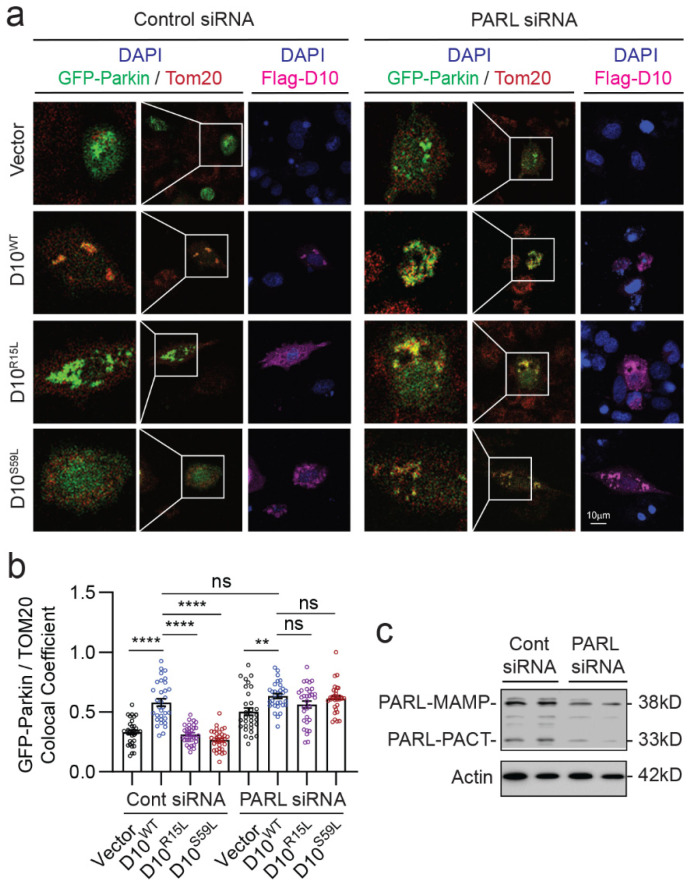
CHCHD10 variants depend on PARL to regulate mitochondrial Parkin recruitment. (**a**) Representative images of HT22 cells transfected with GFP-Parkin (green) and vector control or Flag-CHCHD10 variants (WT, R15L, or S59L) + PARL siRNA, treated with CCCP (1 mM, 12 h), and subjected to staining for Tom20 (red), Flag-CHCHD10 (M2) (far red), and DAPI (blue). (**b**) Quantification of GFP-Parkin colocalization with Tom20 in Flag-CHCHD10 expressed cells (one-way ANOVA, F(7, 247) = 40.96, *p* < 0.0001; posthoc Dunnett, ** *p* < 0.01, **** *p* < 0.0001, ns = not significant, *n* = 32 images from three replicates). (**c**) Representative blots of HT22 cells transfected with control or PARL siRNA and immunoblotted for PARL and Actin.

**Figure 5 cells-12-02781-f005:**
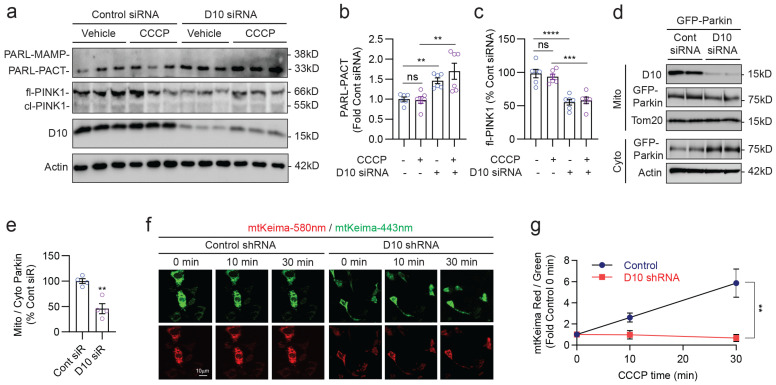
CHCHD10 depletion enhances PARL cleavage and reduces PINK1, suppressing Parkin recruitment and mitophagy flux. (**a**) Representative blots from lysates of HEK293T cells transfected with control or CHCHD10 siRNA, treated with CCCP (1 mM, 12 h), and subjected to immunoblotting for PARL, PINK1, CHCHD10, and Actin. (**b**) Quantification of 33 kDa cleaved PARL-PACT (one-way ANOVA, F(3, 20) = 8.993, *p* = 0.0006; posthoc Sidak, ** *p* < 0.003, ns = not significant, *n* = 6/condition). (**c**) Quantification of fl-PINK1 (one-way ANOVA, F(3, 20) = 20.94; posthoc Sidak, *** *p* < 0.001, **** *p* < 0.0001, *n* = 6/condition). (**d**) Representative blots of mitochondrial and cytosolic fractions isolated from HEK293T cells transfected with GFP-Parkin and control or CHCHD10 siRNA, treated with CCCP (10 mM, 4 h), and subjected to immunoblotting for CHCHD10, GFP-Parkin, Tom20, and Actin. (**e**) Quantification of the mitochondrial to cytosolic GFP-Parkin ratio (*t*-test, ** *p* = 0.0025, *n* = 4/condition). (**f**) Representative live cell images of mtKeima in HT22 cells transduced with control or CHCHD10 siRNA lentivirus and transfected with mtKeima. After CCCP treatment (10 mM) for 0, 10, and 30 min, mtKeima images were detected with dual excitation (580 nm/443 nm -pseudo-colored to red and green, respectively) and 620 nm emission. (**g**) Quantification of the mtKeima-580 nm (red) to mtKeima-443 nm (green) ratio (two-way ANOVA, F(1, 18) = 15.04, ** *p* = 0.0011, *n* = 10 images/condition).

**Figure 6 cells-12-02781-f006:**
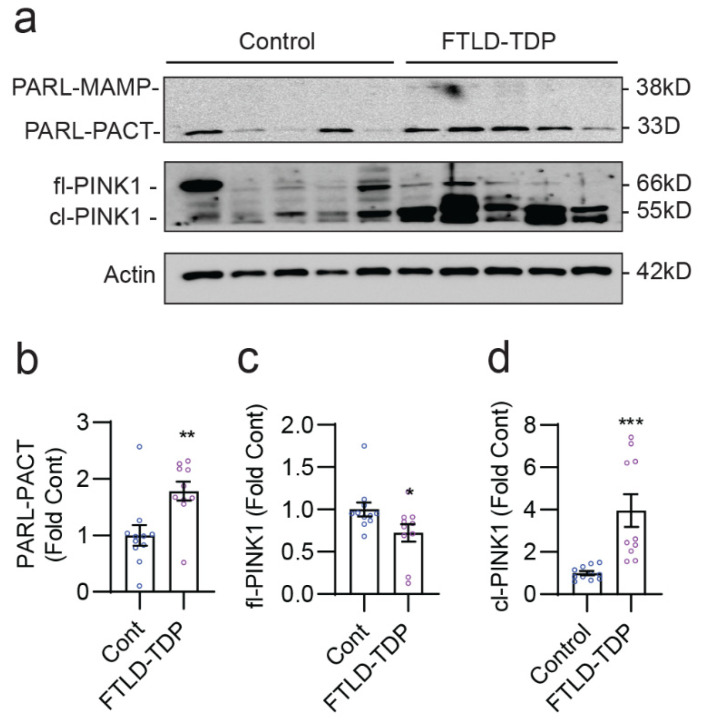
Disruption of the PARL–PINK1 pathway in FTLD-TDP patient brains. (**a**) Representative blots of RIPA-soluble lysates of control and FTLD-TDP frontal gyrus subjected to immunoblotting for PARL, PINK1, and Actin. (**b**–**d**) Quantification of (**b**) 33 kDa cleaved PARL-PACT, (**c**) 66 kDa PINK1 (fl-PINK1), and (**d**) cleaved PINK1 (cl-PINK1) using a *t*-test (* *p* < 0.05, ** *p* < 0.01, *** *p* < 0.001, *n* = 11 control, *n* = 10 FTLD-TDP).

**Figure 7 cells-12-02781-f007:**
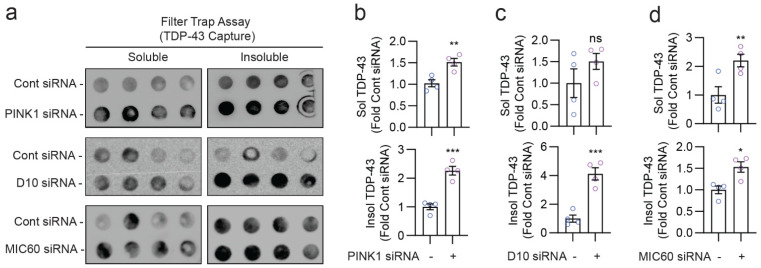
Depletion of PINK1, CHCHD10, or MIC60 promotes TDP-43 aggregation. (**a**) Immunoblots of TDP-43 captured by protein non-binding 0.2 mm cellulose acetate membranes (filter trap assay) using equal protein amounts of RIPA-soluble and RIPA-insoluble lysates of HEK239T cells transfected with TDP-43 ± PINK1 siRNA, CHCHD10 siRNA, or MIC60 siRNA, and treated with CCCP (10 mM, 4 h). (**b**–**d**) Quantification of TDP-43 captured by filter trap assay from RIPA-soluble and RIPA-insoluble fractions (*t*-test, * *p* < 0.05, ** *p* < 0.01, *** *p* < 0.001, ns = not significant, *n* = 4/condition).

## Data Availability

Data supporting the findings of this study are available from the corresponding authors upon reasonable request.

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
