# Peer review of "Disruption of Mitophagy Flux through the PARL-PINK1 Pathway by CHCHD10 Mutations or CHCHD10 Depletion"

_cells, 2023, doi:10.3390/cells12242781_

Round 1

Reviewer 1 Report

Comments and Suggestions for Authors

The study focuses on CHCHD10 and its role in amyotrophic lateral sclerosis-frontotemporal dementia (ALS-FTD). The study also identified mutations (R15L & S59L) in CHCHD10 that impair the process of mitophagy and the recruitment of the protein Parkin to mitochondria. In contrast,CHCHD10WT enhances these processes.

The study further explains that the CHCHD10 mutations (R15L & S59L) reduce the levels of PINK1 by increasing the activity of PARL. CHCHD10WT, on the other hand, suppresses PARL activity by interacting more strongly with it, leading to higher PINK1 levels.

Importantly, the study demonstrates that in the brains of individuals with TDP-43, there is disruption in the PARL-PINK1 pathway. Additionally, when mitophagy is impaired experimentally, it promotes the aggregation of TDP-43. This research provides new insights into how CHCHD10 and the PARL-PINK1 pathway regulate mitophagy and TDP-43 aggregation in the mammalian brain, shedding light on potential mechanisms underlying ALS-FTD.

Comments:

Overall, the manuscript is well written and results are interesting however there are few experimental improvements needed to improve the overall quality of the manuscript.

1)     Figure 1d and e there are no markers are run to determine the purity of the cellular fractionation to determine the cross contamination.

2)     Figure 3 PLA should have a mito tracker or cytoplasmic counter staining.

3)     Many methods needed citations including soluble insoluble fractionation.

Author Response

Dear reviewer,

We would like to thank you for the constructive comments and critiques, which have substantially improved the manuscript. Below, we have addressed all critiques point by point in blue.

1)     Figure 1d and e there are no markers are run to determine the purity of the cellular fractionation to determine the cross contamination.

      Thank you for pointing out this. We have added this data to the supplemental Fig S1.  We used Clpp (mitochondrial matrix protein) as a mitochondrial marker and GAPDH as a cytosolic marker.

2)     Figure 3 PLA should have a mito tracker or cytoplasmic counter staining.

      I greatly appreciate your respectable recommendation. The PLA signals that are not located within the nucleus were quantified. This information has been added to the methods section. The quantification was performed using the intensity per area of cells exhibiting PLA-positive signals that were identical in number. By doing so, bias could be prevented in the absence of mito-tracers or cytoplasmic staining. Additionally, PLA combined with immunocytochemistry may, in our experience, generate more nonspecific imaging signals.

3)     Many methods needed citations including soluble insoluble fractionation.

Thank you for pointing out this. We have added more citations to the methods section.

Reviewer 2 Report

Comments and Suggestions for Authors

The present manuscript demonstrated that CHCHD10 mutations disrupted mitophagy flux through the PARL-PINK1 pathway. It was clearly presented and there were some minor errors.

In Introduction, MIC60 knockdowned in the present study was not explained.

Scale bars were lost in Figs. 1b (high magnification) and 1h.

Were there any differences between two mutations (R15L and S59L) of CHCHD10? In the results, did only Fig. 2b (PARL-MAMP) have a difference between two mutations?

Author Response

Dear reviewer,

We would like to thank you for the constructive comments and critiques, which have substantially improved the manuscript. Below, we have addressed all critiques point by point in blue.

In Introduction, MIC60 knockdowned in the present study was not explained.

Thank you for your suggestion. We have added it to the introduction section (marked red).

Scale bars were lost in Figs. 1b (high magnification) and 1h.

Thank you. We have added the scale bars to these figures.

Were there any differences between two mutations (R15L and S59L) of CHCHD10? In the results, did only Fig. 2b (PARL-MAMP) have a difference between two mutations?

Thank you for your question. We could perceive a pattern in our prior research indicating that CHCHD10 S59L elicits greater neurotoxicity and TDP-43 aggregation than R15L both in vivo and in vitro (PMID: 28585542, PMID: 35787294). Statistically speaking, the effects of the distinction between these two mutations are undetectable. Figure 2b of this study similarly failed to reveal a statistically significant distinction between R15L and S59L.

Round 2

Reviewer 2 Report

Comments and Suggestions for Authors

The authors addressed the reviewer's comments and the manuscript was improved.